# The impact of anthropogenic land use and land cover change on regional climate extremes

Kirsten L. Findell [1], Alexis Berg[2], Pierre Gentine[3], John P. Krasting[1], Benjamin R. Lintner [4], Sergey Malyshev[1], Joseph A. Santanello, Jr.[5] & Elena Shevliakova[1]

Land surface processes modulate the severity of heat waves, droughts, and other extreme events. However, models show contrasting effects of land surface changes on extreme temperatures. Here, we use an earth system model from the Geophysical Fluid Dynamics Laboratory to investigate regional impacts of land use and land cover change on combined extremes of temperature and humidity, namely aridity and moist enthalpy, quantities central to human physiological experience of near-surface climate. The model's near-surface temperature response to deforestation is consistent with recent observations, and conversion of mid-latitude natural forests to cropland and pastures is accompanied by an increase in the occurrence of hot-dry summers from once-in-a-decade to every 2–3 years. In the tropics, long time-scale oceanic variability precludes determination of how much of a small, but significant, increase in moist enthalpy throughout the year stems from the model's novel representation of historical patterns of wood harvesting, shifting cultivation, and regrowth of secondary vegetation and how much is forced by internal variability within the tropical oceans.

[1] Geophysical Fluid Dynamics Laboratory, 201 Forrestal Road, Princeton, NJ 08540, USA. [2] Princeton University, Department of Civil and Environmental Engineering, Princeton, NJ 08544, USA. [3] Columbia University, Department of Earth and Environmental Engineering, 918 S.W. Mudd Hall, Mail Code 471, 1500 West 120th Street, New York, NY 10027, USA. [4] Rutgers University, Dept. of Environmental Sciences, 250 Environmental & Natural Resource Sciences Building, 14 College Farm Road, New Brunswick, NJ 08901-8551, USA. [5] NASA GSFC Hydrological Sciences Branch, Mail Code 617, Greenbelt, MD 20771, USA. Correspondence and requests for materials should be addressed to K.L.F. (email: Kirsten.Findell@noaa.gov)

The catastrophic summertime heat wave events in Western Europe in 2003 and Russia in 2010 inspired much new research on extreme events (e.g., refs [1–3]). Vegetation and surface moisture conditions have been shown to impact both the severity and duration of heat wave events (e.g., refs [4–7]), as well as the aridity over land in the future[8]. In particular, it has been shown that different intensities of plant water consumption (from more intensive to more conservative water use) can drastically impact heat wave severity[9–11].

Previous model-based studies of the impact of land use and land cover change (LULCC) on temperature extremes have produced contrasting results[12]. For instance, ref. [13] shows that daily temperature extremes are less severe with deforestation, while ref. [14] shows enhanced severity of June extremes. This spread of model response in extreme temperatures is consistent with the more extensive literature related to the mean temperature response to LULCC. In the temperate mid-latitudes (where most of historical LULCC has taken place), there is uncertainty about the relative importance of competing biophysical influences[15, 16], with some models showing that the albedo-driven warming effect of forest cover is dominant (e.g., refs [17, 18]), and others showing that deeper roots and higher turbulent exchange of forests yield cooler conditions than pastures or crops (e.g., refs [19–22]). Recent work has also demonstrated that surface roughness differences are the dominant factor in the Amazon[23, 24] and in the northern mid-latitudes[25]. Overall, the LUCID (Land Use and Climate: IDentification of robust impacts) intercomparison project[15] demonstrated that six out of the seven contributing climate models show summertime cooling in the mid-latitudes as a result of historical LULCC, with little temperature response in the tropics (though ref. [26] showed that in South America dry season temperature and precipitation are both significantly impacted by LULCC). LUCID results were found to be qualitatively similar for temperature extremes, although for a more limited set of models, with three out of four models showing cooler extremes with deforestation[12].

Recent observational studies have shed some light on the climatic effects of land use and land cover differences. Observations support a latitudinal dependence in the balance between the cooling and warming factors influencing the response to deforestation[27]. However, contrary to many climate models (e.g., ref. [15]), satellite-based observations show that non-radiative mechanisms dominate the local surface air temperature response to land cover change[28], and that cleared lands in the mid-latitudes are warmer than nearby forests in both daily mean and maximum air temperatures, particularly in summer but also in the annual mean[28–30]. Similarly, surface air temperatures in mid-latitude forests throughout the US are cooler than nearby cleared lands from spring through fall, and in all seasons in the southeastern US[31].

While most previous studies of LULCC impacts have focused on single-variable assessments typically involving temperature and precipitation (e.g., refs [15, 19, 20]), we assess the impact of anthropogenic LULCC on regional climate extremes using a novel approach to characterize the joint temperature–humidity response to LULCC. Ref. [32] argued for use of surface air heat content, including both temperature and humidity, as a more complete measure of global change than temperature alone. Additional studies have considered both the temperature and humidity of near-surface air to calculate quantities relevant to the human capacity to cope with or adapt to climatic conditions (e.g., refs [33–38]). Furthermore, plant physiological stress and ecosystem function are heavily impacted by the vapor pressure deficit of near-surface air[39]. Including both humidity and temperature in this analysis allows us to consider aridity and moist enthalpy: quantities that are closely related to vapor pressure deficit and wet

bulb temperature, respectively, and are central to ecosystem health and human physiological perception and experience of near-surface climate conditions[35].

Here, we compare historical (i.e., 1861–2005) all-forcing simulations (AllHist) of the model GFDL-ESM2G (part of the Coupled Model Intercomparison Project CMIP5;[40]) to simulations without the historical land use reconstruction[41], that is, with only potential vegetation through the entire simulation (PotVeg; the vegetation that would be present at each grid cell with no human interference in the landscape). We demonstrate that GFDL-ESM2G displays a monthly mean temperature sensitivity to land cover conversion that is consistent with available observations[29–31] in boreal, temperate, and tropical biomes. In the mid-latitudes, the PotVeg simulation produces hot-dry summers far less frequently than the AllHist simulation. The consistency with observational constraints is a critical factor providing important rationale for our analysis. GFDL-ESM2G is at the forefront of characterizing land use processes, such as wood harvesting on primary and secondary lands, shifting cultivation in the tropics, and secondary vegetation regrowth[42]. These processes are particularly important in the tropics, though the long time-scale of tropical oceanic variability complicates interpretation of tropical results. Although we clearly acknowledge the limitations of a single model in terms of generalizability, the analysis protocol introduced here can be applied to output from the suite of models participating in the upcoming Land Use Model Intercomparison Project (LUMIP;[43]).

## Results

**Anthropogenic land use and land cover change**. Figure 1a shows the spatial coverage of natural vegetation depicted in the PotVeg run, while Fig. 1b–e show the average fraction of the four land-use types at the end of the AllHist experiment (averaged over the last 25 years of each of two AllHist simulations). Secondary vegetation is prominent throughout Europe, along the southern border of the cold evergreen forests of Eurasia, in the eastern and coastal northwestern United States, around the margins of Amazonia, and in large sections of northern and central Africa. National boundaries are clearly evident in the natural and secondary characterizations of northern Africa as a result of using national wood harvest statistics as an input to the land use reconstruction[41].

Croplands are prominent in the upper Midwestern US and extending into the central provinces of Canada, in India, particularly in the Ganges River basin, and throughout Europe, particularly eastern Europe. More modest percentages of croplands are evident in Central America, the eastern part of West Africa, and in southeastern China. Pastures are dominant in the central US, throughout central Asia, the Sahel, southern Africa, southeastern South America, and Australia.

**Model response to deforestation compared with observations**. Given the differing model responses to mid-latitude deforestation discussed above, we first evaluate the effect of land use on surface climate in GFDL-ESM2G against recently available temperature observations from ref. [29] (hereafter AC16) covering the decade from 2003 to 2012. AC16 uses satellite-derived estimates of temperature within 0.05° resolution grid cells that are considerably smaller than the 2° latitude by 2.5° longitude grid resolution in the simulations analyzed here. Thus, we use sub-grid scale tile-specific monthly mean canopy air temperature differences between crop-covered tiles and natural forest-covered tiles within the same grid cell as the best modeled proxy for the surface air temperature differences due to forest cover loss documented by AC16.

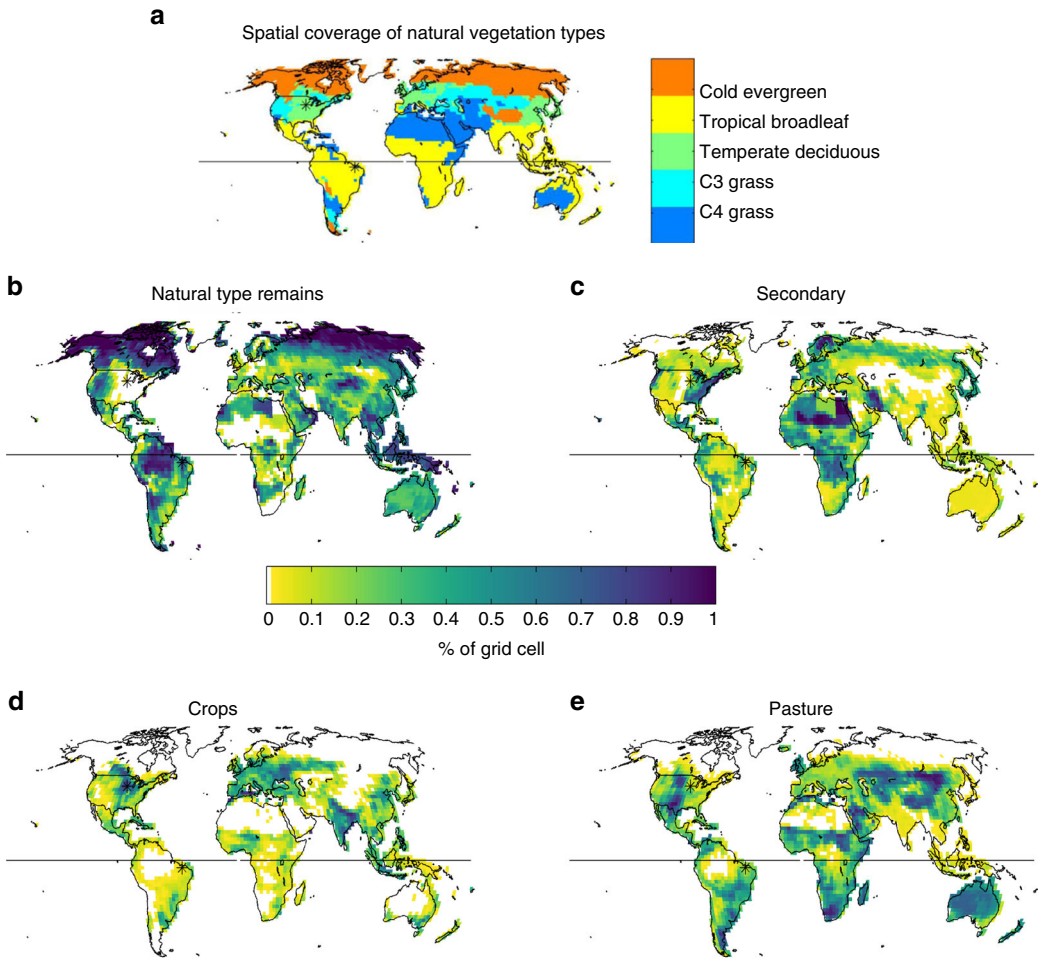

**Fig. 1** Potential vegetation cover and anthropogenic land cover conversions. **a** Natural vegetation types. Percent of land remaining as **b** natural type, or converted to **c** secondary, **d** crops, and **e** pasture. **b**–**e** Averaged over 1981–2005 in two ensemble members of the AllHist simulations. *Black asterisk* over points in the central United States and Eastern Brazil

Figure 2 shows the seasonal cycle of these differences in canopy air temperature ($T\mathrm{ca}_{crop} - T\mathrm{ca}_{natural\ forest}$) for evergreen coniferous, temperate deciduous, and tropical broadleaf forests compared with AC16's observation-driven estimates of seasonal changes in surface air temperature due to forest losses in boreal, temperate, and tropical climate zones. Supplementary Fig. 1 provides maps of the model's seasonal differences in canopy air temperature where those differences are statistically significant. Ref. [22] obtain similar results when comparing sub-grid temperature differences between crop and forest tiles in the GFDL-ESM2Mb model, which shares the same land and atmosphere components as ESM2G here (but a different ocean component, see ref. [44]). The model's response to deforestation in each climate zone is broadly consistent with the observations (Fig. 2). In the boreal zone in northern hemisphere (NH) winter, the albedo effect dominates, producing a cooling signal with deforestation in both the model and the observations. This persists through May in the model mean—one month longer than in the observations, though the observations are always within the broad error bars, indicating that there is substantial variability over the boreal grid cells and the 50 years analyzed.

In the temperate zone, the observations show a clear warming with deforestation throughout the year, peaking in NH summer. The annual cycle of the model's response is consistent with the AC16 observations, though the magnitude of the difference is too small, particularly in the second half of the year, and the late fall

and early winter sign reversal is not consistent with the observations. Nevertheless, Fig. 2 demonstrates that the albedo-driven processes dominant in boreal winter do not drive the mid-latitude response. These data are consistent with the models which simulate regional warming in response to mid-latitude deforestation[19, 20, 22].

In the tropics, the observations show warming with deforestation of about 1 °C, with little seasonality. The magnitude of the model response is comparable to the observations at the beginning of the year after the annual crop harvesting is applied, but the model's signal is attenuated later in the year, while the observational difference is relatively constant throughout the year. Future model development is slated to include more realistic crop harvesting methodology. Overall, Fig. 2 shows that GFDL-ESM2G is in qualitative agreement with recent observations regarding the climatic impacts of land cover differences, both in terms of latitudinal and seasonal effects.

**Northern mid-latitude response to anthropogenic LULCC.** To analyze the effect of anthropogenic LULCC on global surface climate, comparisons are made between 50 simulated model years of data for each experiment. Statistical significance is determined through application of two-tailed modified *t*-tests at the 90% significance level for Figs. 3 and 4. The modified *t*-test accounts for autocorrelation within the time series[45–47].

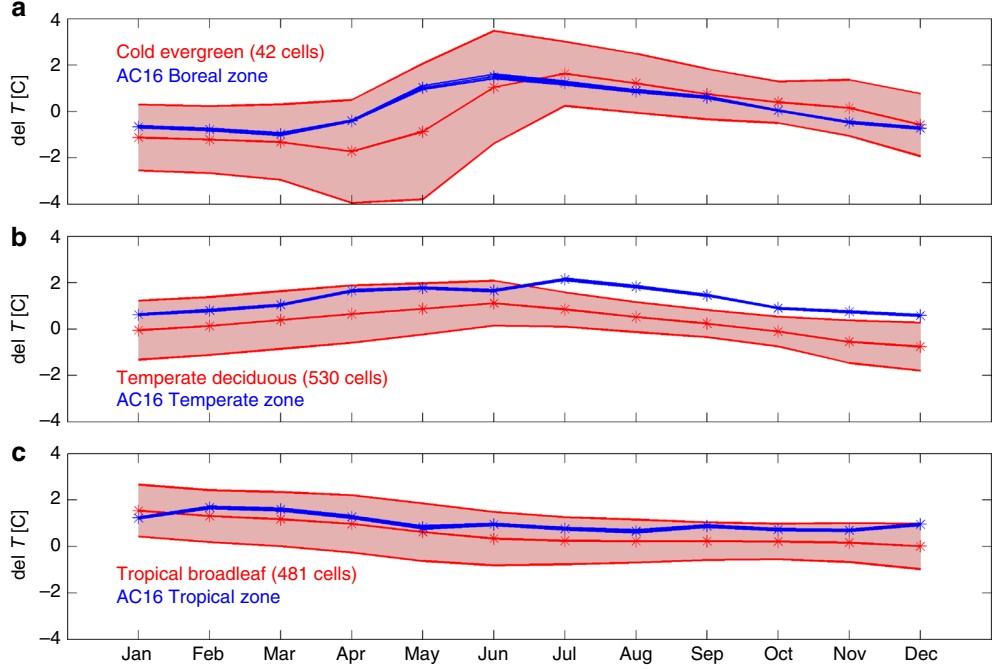

**Fig. 2** Temperature change in response to deforestation in model and observations. *Red lines*: ESM2G results. Mean differences between the canopy air temperature [°C] in crop covered tiles and tiles with natural vegetation within the same grid cell. For grid cells where at least 5% of the grid cell area was converted to crops, and the natural vegetation was **a** cold evergreen forests; **b** temperate deciduous forests; and **c** tropical broadleaf forests. *Shaded regions* are ±1σ of the differences at all years (two AllHist runs, 1981–2005) at all grid cells within each zone. *Blue lines*: Alkama and Cescatti's (2016[29]) observation-driven estimates of seasonal changes in surface air temperature due to losses of forest cover (as presented in their Fig. 2a) for the **a** boreal, **b** temperate, and **c** tropical climate zones defined in their paper. *y*-axis value is temperature sensitivity [°C] to 100% deforestation within a 0.05° grid cell. (Printed with permission of Alkama and Cescatti)

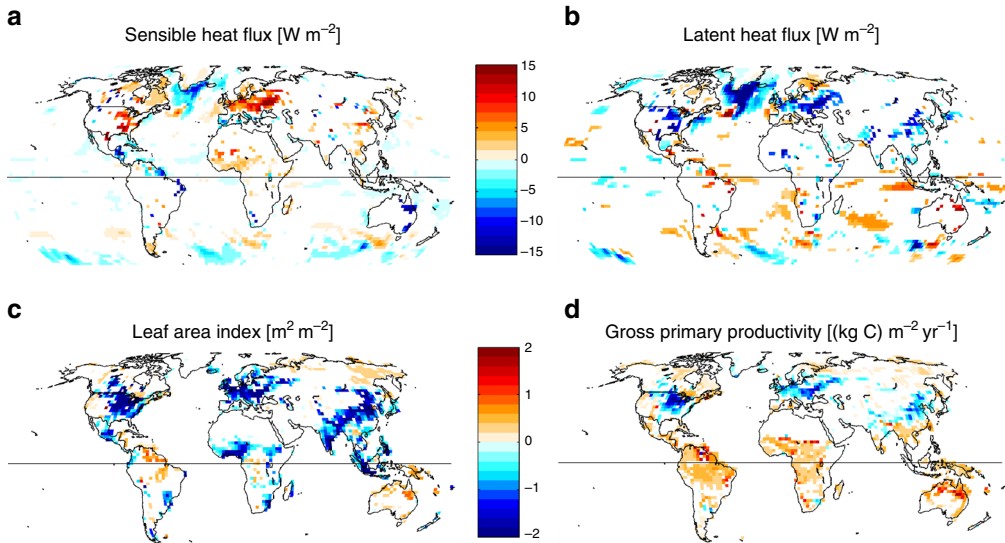

**Fig. 3** Mean model response to anthropogenic LULCC. Mean differences between the AllHist and the PotVeg simulations for June–August (JJA) in the northern hemisphere and for December–February (DJF) for the southern hemisphere, averaged over the years 1981–2005 in two ensemble members for each experiment. **a** Sensible heat flux; **b** latent heat flux; **c** leaf area index; and **d** gross primary productivity. Differences shown pass a modified *t*-test at the 90% significance level

The summertime northern mid-latitude response to anthropogenic LULCC (Fig. 3) is dominated by a reduction of latent heat flux (particularly through transpiration, not shown) and an increase of sensible heat flux in the altered regions, particularly in regions converted to croplands. These areas also show large reductions in leaf area index (LAI) and gross primary productivity (GPP). As shown in Fig. 4 in combined humidity-temperature phase-space (see methods), the changes in vegetation characteristics and functioning are accompanied by a statistically significant mid-latitude warming and drying of the near-surface atmosphere, i.e., the differences pass a modified *t*-test accounting for autocorrelation within the time series[45–47]. These results are consistent with the analysis of temperature fields in refs [19, 20] using earlier generation GFDL models, and in ref. [22] using the

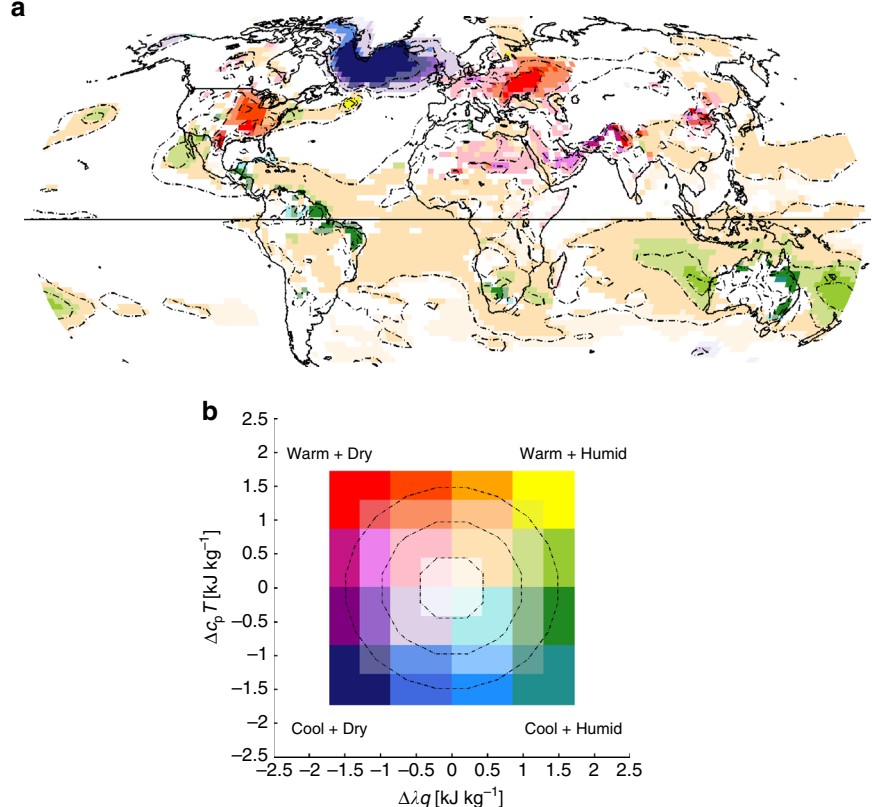

**Fig. 4** Joint temperature-humidity response to anthropogenic LULCC. **a** Mean differences (AllHist–PotVeg) for the time period 1981–2005 for June–August (JJA) in the northern hemisphere and for December–February (DJF) for the southern hemisphere in ($\lambda q$, $c_p T$) phase space; differences of 2 m values of $\lambda q$ and $c_p T$ are plotted according to the two-dimensional colorgrid shown in panel **b**. $q$ is specific humidity, $\lambda$ is the latent heat of condensation, $T$ is temperature, and $c_p$ is the specific heat of dry air at constant pressure. Colors reflect the vector differences of the means when plotted on conserved variable diagrams. Differences shown have at least one of the two variables passing a modified $t$-test at the 90% significance level. Contours every 0.5 kJ kg$^{-1}$

GFDL-ESM2Mb model: in contrast with many other climate models[15], the GFDL models all show statistically significant near-surface warming in regions with historical LULCC, in qualitative agreement with recent observations, as discussed above (Fig. 2). Furthermore, ref. [20] showed that in regions subjected to substantial LULCC, changes in climatic conditions were similar in magnitude to those occurring in response to realistic sea surface temperature anomalies and to greenhouse warming during the 20th century.

The North Atlantic region stands out in Figs. 3 and 4, with substantially cooler and drier conditions in the AllHist runs in the sub-polar gyre (with a small band of warmer waters to the south), a pattern reminiscent of the impact of the Atlantic Meridional Overturning Circulation (AMOC) variability on ocean temperatures in ESM2G, discussed in ref. [48]. The AMOC exhibits multi-decadal variability on time-scales greater than 30 years in many climate models[48]. In ESM2G, the maximum multi-decadal AMOC spectral power has a period of 40 years[48], suggesting that 25-year blocks like those analyzed here could be locked in different AMOC states. As a result, the North Atlantic changes seen here cannot be attributed to LULCC-driven changes in climate: a robust characterization of the differences in this region would require far more data than two 25-year periods. Indeed, analysis of five 20-year long integrations with ESM2Mb presented in ref. [22] indicates warming in the North Atlantic. Furthermore, Supplementary Figs. 2 and 3 show no significant differences in this region during the 1861–1885 time period of the ESM2G runs. The lack of consistency between these results suggests that the signal in the North Atlantic in Figs. 3 and 4 is not robust due to

long time-scale internal variability of the ocean, and may be model-dependent.

The mixing diagrams of Fig. 5 illustrate how LULCC modifies diurnal cycle behavior (see methods for details). July monthly mean diurnal cycles (7:00 a.m. to 6:00 a.m.) for the 50 years of each scenario (1981–2005 for each of two ensembles members) are shown in Fig. 5a, b for a single grid point in Iowa, which is broadly representative of midlatitude croplands. Over the sampled pixel, nighttime values tend to be at or near saturation for the PotVeg diurnal cycles (Fig. 5b), while the mid- to late-afternoon values generally have relative humidity (RH) values of 75–85% (Nighttime super-saturation in Fig. 5b results from extrapolation from the lowest model level down to 2 m during post-processing. All results presented here are consistent with calculations performed with mean values from the lowest model level or from 925 mb (not shown), but diurnal cycles were only saved for 2 m values). The 50 corresponding AllHist July diurnal cycles (Fig. 5a) show much warmer and drier conditions, with no instances of afternoon RH values above 80%, and a few years falling below 50% RH in the late afternoon. The mean diurnal cycles for these 50 years (Fig. 5c) reflect these changes: anthropogenic LULCC shifts the mean July diurnal cycle in Iowa to warmer, drier conditions, with afternoon RH decreasing from close to 80% in the PotVeg experiment to <70% in the AllHist experiment.

The shift in the daily mean of the mean diurnal cycles shown in Fig. 5c is the value plotted at this location in Central Iowa in Fig. 4. The small scale of the mean differences shown in Fig. 4 (order 0.5% for $c_p T$ and 5% for $\lambda q$) reveals nothing about the

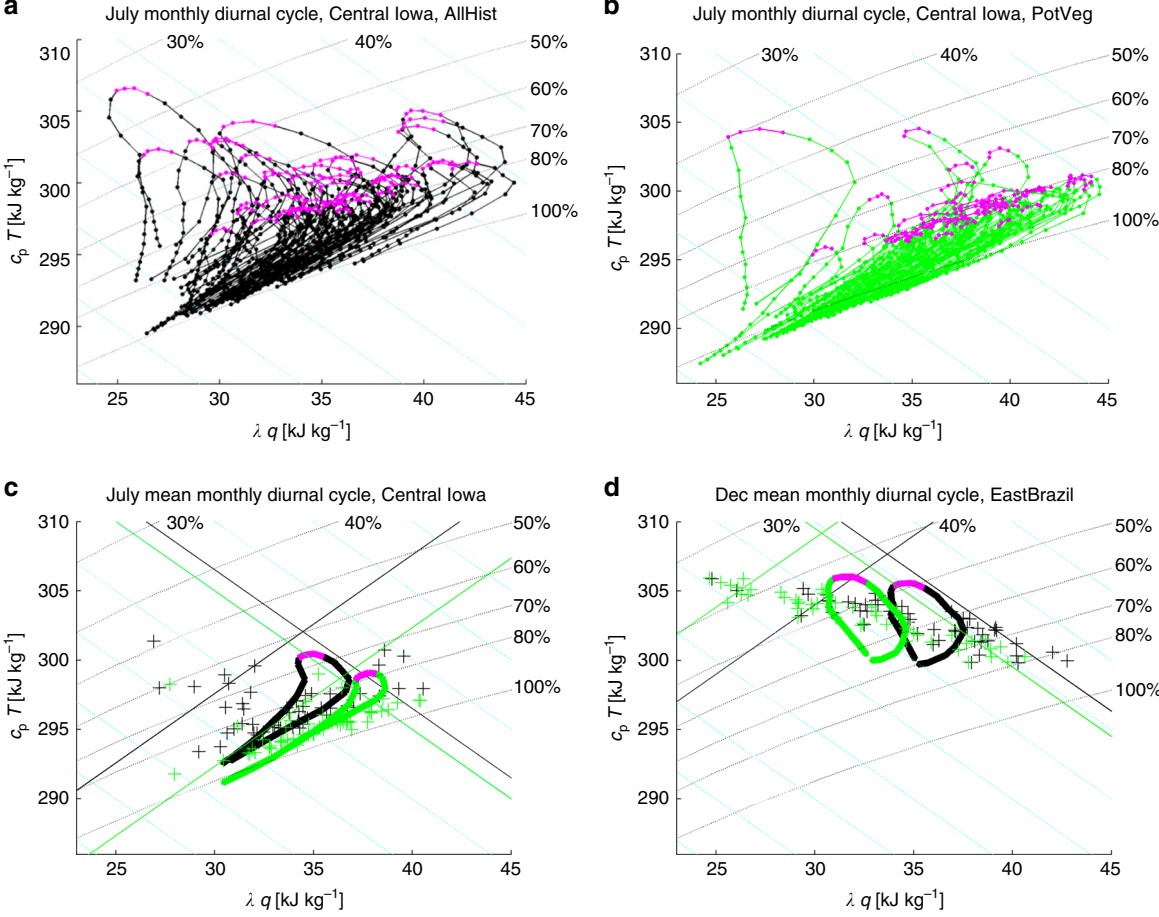

**Fig. 5** Change in diurnal cycle variability at representative grid points in response to anthropogenic LULCC. **a**, **b** For a single grid point in Central Iowa, AllHist **a** and PotVeg **b** runs, two ensemble members each, 1981–2005 (50 years total), 7:00 a.m.–6:00 a.m., mean monthly diurnal cycles of 2 m values of $\lambda q$ and $c_p T$ for July plotted in conserved-variable space typically used for mixing diagrams. $q$ is specific humidity, $\lambda$ is the latent heat of condensation, $T$ is temperature, and $c_p$ is the specific heat of dry air at constant pressure. *Magenta symbols* for 3–6 pm values. **c** *Thick lines*: mean diurnal cycles from **a**, **b**; *magenta lines* from 3–6 pm; *plus signs*: mean July $(\lambda q, c_p T)$ for each of the 50 years. **d** As in **c**, but for a grid cell in East Brazil for December. In all plots, *dashed black lines*: constant relative humidity; *dashed cyan lines*: constant moist enthalpy (ME). In **c**, **d** *solid lines* are thresholds for the 90th percentile monthly mean values of ME and HD (see methods). In all plots, *green* is used for the PotVeg runs, *black* for the AllHist runs

change in extremes shown in the comparison of Fig. 5a, b. The natural deciduous forest cover of the PotVeg scenario leads to fewer excursions into the upper left hot, dry portion of the $(\lambda q, c_p T)$ phase-space (Fig. 5b), indicating that natural deciduous forests modulate the occurrence of extremely hot, dry summer-time conditions in Iowa, relative to the crop-covered historical land use scenario (Fig. 5a). Figure 5a, b also demonstrate that in most years the mean July diurnal cycle of temperature and humidity is much more tightly constrained over forests than over croplands. These results are consistent with ref. [9] who found that forests mitigate the impact of the most extreme heatwave events in Europe.

To quantify the frequency and extent of the tempering of hot, dry extremes by vegetation, we wish to compare how frequently the data from each experiment occupy the most arid (*upper left*) portion of the mixing diagrams in Fig. 5. This is the portion of the parameter space with the highest hot-dry (HD; see methods) values. We calculate the 90th percentile threshold values for the monthly mean HD, as well as the 90th percentile threshold values for moist enthalpy (ME; see methods). These threshold values are shown in Fig. 5c with solid green (PotVeg) and black (AllHist) lines, with each line showing 5 of the 50 years at or beyond the experiment's threshold value. For example, the 90th-percentile HD threshold for the AllHist experiment at the grid cell in

Central Iowa is shown by the constant HD black line running from the lower left to the upper right of Fig. 5c. The equivalent threshold for the PotVeg experiment is about 5 kJ kg$^{-1}$ smaller (parallel green line), indicating that the threshold for a 10-year return period hot, dry event in the AllHist scenario is substantially hotter and drier than it would have been without anthropogenic LULCC. Figure 5c shows that about half of the July values in the AllHist run exceed the 90th percentile threshold for hot, dry conditions determined from the PotVeg run: in other words, what is a once-in-a-decade hot, dry summer in the PotVeg scenario is an every-other-year occurrence in the AllHist scenario at this grid point.

Figure 6a underscores that this shift in hot, dry summer return interval is not limited to Iowa, and is not limited to July. Based on these simulations, the conversion of forests to cropland is coincident with much of the upper central US and central Europe experiencing extreme hot, dry summers (as defined by the PotVeg scenario) every 2–3 years instead of every 10 years. This signal even emerges in the seasonal zonal mean, with an average doubling of the frequency of these extreme hot, dry summers at 50°N resulting from anthropogenic LULCC (Fig. 6b). Note also that the Wilcoxon rank-sum test used in Fig. 6 is conservative, in that the significance test for each pixel is based on differences in the mean only. In other words, the significance test does not

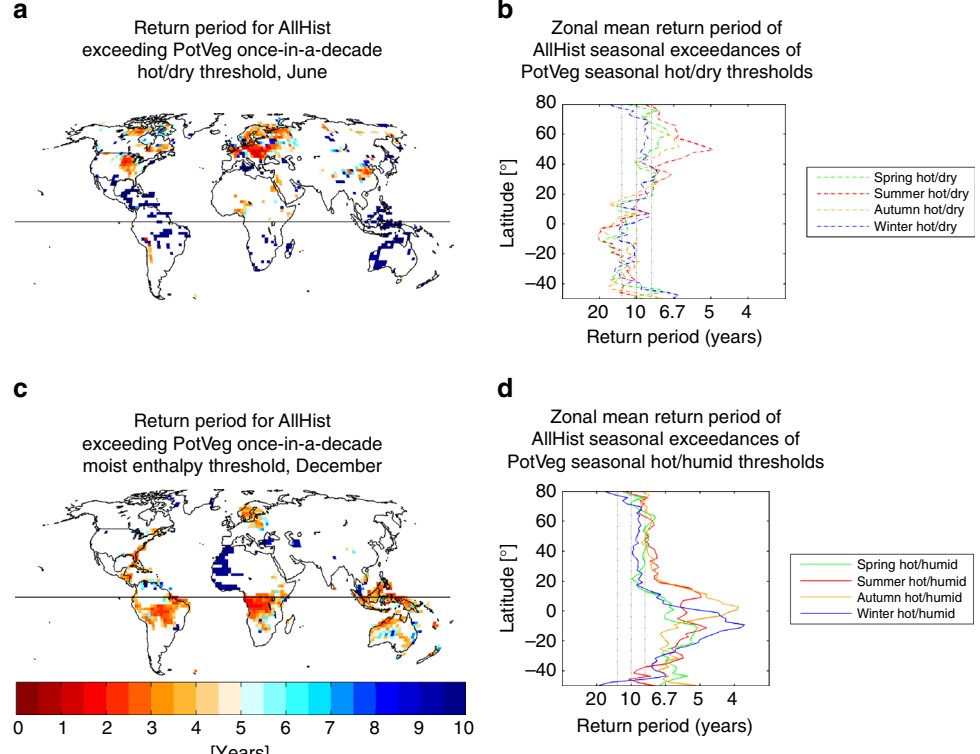

**Fig. 6** Change in return periods of extreme events in response to anthropogenic LULCC. AllHist return periods for exceedance of the 90th percentile (once-in-a-decade) event thresholds determined from the PotVeg run for the 1981–2005 time period for **a** the hot-dry threshold in June, **c** the moist enthalpy threshold in December, and seasonal, latitudinal means for **b** hot-dry thresholds, and **d** moist enthalpy thresholds. The *colorbar* is for both **a**, **c**, and colors are only shown where the medians of the 50-year samples are significantly different according to a Wilcoxon rank sum test at the 5% level. In **b**, **d** *gray vertical lines* are drawn at 6, 10, and 14-year return periods (5 ± 2 exceedances in 50 years), with 10 years representing the null hypothesis (no change from PotVeg). In the mid-latitudes, summertime aridity exceeds these bounds over a ~40 degree latitude band. In the tropics, all seasons demonstrate more frequent high moist enthalpy conditions

identify pixels for which differences in the distribution tails may occur with little to no shift in the mean of the distribution—a behavior that can arise as a result of land-atmosphere feedbacks[49].

While the results presented here emphasize monthly daily means, we have similarly analyzed the mean afternoon conditions (3–6 pm, *magenta points* in Fig. 5; 3–6 pm typically captures the maximum temperature and includes the sometimes dramatic afternoon drying which often occurs around the time of peak temperature) as well as the maximum daily temperature and the associated humidity. We find comparable results, with seasonally averaged extreme hot, dry afternoon conditions occurring every 2–3 summers instead of every 10th summer as a result of anthropogenic conversion from mid-latitude forests to crops and pastures (not shown). Some of this similarity stems from the use of monthly mean diurnal cycles rather than values from hourly data saved every day—a limitation imposed by data availability. Although the magnitude of the change of extreme occurrence return period is likely to depend on both the model analyzed and assumptions about LULCC history, the size of the shifts we find for GFDL-ESM2G points to a significant role for LULCC in shaping the distribution of regional extremes in the current climate.

Supplementary Figures 2–4 are equivalent to Figs. 3, 4, and 6, but for the period 1861–1885. The pattern of differences in the mid-latitude mean fields over land are generally similar in the late 1800s to those at the end of the 20th Century, but over a reduced geographical area with far fewer significant pixels and with differences of smaller magnitude (Supplementary Figs. 2 and 3).

Shifts in extremes as quantified in Fig. 6 are absent in the late 1800s (Supplementary Fig. 4). Thus, it is only with LULCC change during the more recent historical period that return periods of extremes are appreciably altered.

**Tropical response to anthropogenic LULCC.** Tropical land regions with intensive crop development (India, parts of SE Asia, part of West Africa; Fig. 1) manifest decreasing LAI, as in the mid-latitudes (Fig. 3c). However, unlike the mid-latitudes, tropical land regions have a substantial fraction of primary or secondary forests, and show, on average, increased GPP relative to the PotVeg scenario (Fig. 3d), mostly without significant changes in sensible or latent heat flux (Fig. 3a, b).

On the other hand, isolated pixels show LAI increases in three tropical broadleaf forest regions: (1) the Amazon, with ~10% pasture around the margin of the forest region and ~10% secondary throughout; (2) central Africa, with a strong imprint of secondary vegetation; and (3) northern Australia, with about 50% replacement by pasture and ~10% secondary (Figs. 1 and 3c). Under the LULCC scenario assumptions in the reconstruction[41], shifting cultivation is represented by 6.7% of croplands and pastures abandoned annually (i.e., converted to secondary) and an equal area of new croplands and pastures being created by conversion of natural or secondary lands. Previous studies have highlighted the importance of secondary vegetation and indicated that shifting cultivation practices in tropical regions significantly influence carbon dynamics[50, 51]. Because of their effect on vegetation characteristics, such practices could also modulate biophysical feedbacks. However, the LAI increase in Amazonia

and tropical Africa is not consistent with the LAI response in GFDL-ESM2Mb used in ref. [22], although both models share the same vegetation scheme. Indeed, further analysis of the five ensemble members of the runs used in that earlier study indicates that different combinations of two ensemble members in each scenario can yield positive or negative LAI differences in the Amazon (not shown), suggesting that these LAI changes are not robust despite the fact that some isolated pixels do pass the modified *t*-test applied in Fig. 3.

The tropical oceanic response to anthropogenic LULCC (Fig. 4) also differs in the two GFDL ESMs. The slightly warmer and more humid conditions over the tropical Atlantic, the Indian, and the West Pacific in the AllHist mean shown in Fig. 4 are not replicated in the ESM2Mb runs; in fact, ref. [22]'s Fig. 11 only exhibits a significant signal in the East Pacific, where there is slight warming. Furthermore, Supplementary Fig. 3 demonstrates that in the 1861–1885 time period, the ESM2G AllHist mean is slightly cooler and less humid over portions of the East Pacific and southern Atlantic. Observational analysis and modeling studies suggest that sea surface temperatures in the western tropical Pacific show predominant variability over multidecadal time scales, and that this variability is linked to the Atlantic meridional oscillation through atmospheric bridge-style mechanisms[52, 53]. Similar to the model oceanic response over the North Atlantic discussed above, the contrasting results between ESM2G and ESM2Mb over the tropical oceans all point to long time-scale variability in the ocean and insufficient sampling of that oceanic variability. The persistence of increased tropical moist enthalpy in all seasons shown in Fig. 6d also suggests long time-scale oceanic processes rather than vegetation-driven changes, which would be expected to show more seasonality.

Focusing on the tropical land regions with substantial LULCC, e.g., the semi-arid Nordeste region of Brazil, highlights the shift towards higher humidity and correspondingly higher moist enthalpy (Fig. 5d). This increase is consistent with an increase in LAI, GPP, and latent heat flux over that region (Fig. 3), and suggests a possible local vegetation-forced response, but the signals are only significant over a few isolated grid cells. A similar response is visible over northeast Australia. Furthermore, Fig. 6d shows that higher ME is a year-round signal throughout the tropics: each of the tropical land regions manifests a twofold to threefold increase in the frequency of high humidity events in all seasons (Fig. 6c, d). As discussed above, it is not clear how much of this signal is forced by internal variability of the ocean and how much is forced by the LULCC scenario, or by interactions between the two. This tropics-wide temperature and humidity response (Fig. 4) is likely linked to the increased GPP throughout the tropics (Fig. 3d) and is consistent with the small but significant tropics-wide upper-level temperature and geopotential height response to large-scale deforestation experiments discussed in ref. [47] using an earlier generation GFDL model. Studies focusing on only the temperature effect of LULCC may not capture the significant humidity-driven changes in near-surface tropical climate highlighted here. Given the physiological importance of moist enthalpy, or similarly wet bulb temperature, for humans and mammals[35] and the rapid rise in urbanization in tropical regions, our results underscore the importance of LULCC as a necessary component in any characterization of future climate conditions and heat stress in tropical regions. However, given the long time scale of oceanic variability, longer simulations with more ensemble members are required to reduce the possibility of sampling only a subset of distinct low-frequency oceanic modes of variability and to attribute the climate response to land-use processes.

## Discussion

This study lends a new perspective on the role of anthropogenic LULCC on regional climate extremes, both through the novel consideration of the joint temperature–humidity response to LULCC and through the use of an ESM, which captures effects of sub-grid-scale deforestation and accounts for critical processes such as wood harvesting, shifting cultivation, and secondary vegetation growth. We demonstrate that GFDL-ESM2G's sub-gridscale mean temperature responses to deforestation are consistent with the most recent and comprehensive observations in boreal, temperate, and tropical ecosystems. The inclusion of anthropogenic LULCC within this model produces a twofold to fourfold increase in the frequency of hot, dry summers over much of the mid-latitudes compared to simulations with potential vegetation. In other words, as a result of anthropogenic LULCC, the threshold value for what we identify as a HD summer in the mid-latitudes is substantially hotter and drier than it would have been without these alterations to the land surface. This mid-latitude response is consistent with earlier assessments of changes in summertime mean temperatures in response to anthropogenic LULCC (e.g., refs [19, 20]), but here we demonstrate that these changes in mean temperatures extend to changes in extremes of temperature and humidity. Given the importance of both temperature and humidity for the health and well-being of humans and ecosystems, this bivariate approach enhances understanding of the broader impact of LULCC.

The tropical response in this model appears to be strongly impacted by low-frequency oceanic variability that is not adequately sampled with the limited number of simulated years available here. The tropical land response further hinges on the inclusion of secondary forest growth, wood harvesting, and shifting cultivation in GFDL-ESM2G—processes not included in many CMIP5 ESMs. Refs [50, 51] showed that these practices impact carbon dynamics; here we demonstrate that they may also impact the tropical hydrologic cycle, though longer simulations are needed to fully assess this impact. Nevertheless, the results presented here support the need for further study and echo calls of previous research (e.g., refs [16, 43, 54, 55]) calling for high resolution representations of LULCC in Earth System model projections of future climate.

## Methods

**Model and experiments**. Model data analyzed are from two sets of simulations of the GFDL-ESM2G model[42, 51, 56], which includes the terrestrial component LM3, representing both vegetation dynamics[50] and land hydrology[57]. The fully coupled ocean and atmospheric component models are described in ref. [51]. The atmosphere and land components' horizontal grid increment is 2° latitude by 2.5° longitude, with 24 vertical levels in the atmosphere and 20 layers in the soil. LM3 simulates changes in vegetation and soil carbon pools, effects of LULCC on them, as well as exchanges of water, energy, and carbon between the land and the atmosphere. Vegetation is represented by one of five types: deciduous, coniferous, and tropical trees, as well as cold and warm grasses. The biogeography parameterization uses the total biomass in a tile in combination with prevailing climatic conditions to determine the vegetation type. Subgrid land-use heterogeneity is represented by a collection of natural (i.e., potential vegetation) and up to 12 different anthropogenically disturbed tiles (i.e., cropland, pastures, and secondary). The model simulates in each tile above- and below-ground hydrology, energy balance, and vegetation characteristics such as vegetation height, LAI, and albedo.

The changes in the sub-grid land-use composition are prescribed annually from the CMIP5 land-use reconstruction[41] for each grid cell in terms of transition rates between four different land-use types: natural (i.e., undisturbed by humans), croplands, pastures, and secondary lands (i.e., previously logged or abandoned). The transition-based approach used in LM3 creates more land-cover disturbance than the fraction-based approach because the transitions reflect the paths of changes among different use categories—and thus the gross transitions—between different land-use types, rather than just the net effect based on changes in fractions[42]. These gross sub-grid changes include shifting cultivation and secondary-to-secondary transitions representing wood harvesting of secondary forests.

Each grid cell can have up to ten secondary tiles to capture age and thus biomass distribution of recovering lands. As vegetation in a secondary tile grows

and ages and its biomass becomes similar to that in older tiles, the model merges them to avoid proliferation of tiles and to increase computational efficiency. Harvesting of crops and pastures is applied annually, though with differing intensities: all but 0.1 kg C m$^{-2}$ of crop biomass is removed annually, while on pastures, 25% of leaf biomass is removed each year[50]. Some grid cells also include lakes and/or glacier tiles.

On vegetated tiles evapotranspiration in LM3 can occur through three pathways: from soil and/or snow surface evaporation, plant interception, and through transpiration. Transpiration is a function of plant stomatal conductance and soil water availability, which depends on the vertical distribution of plant roots and soil moisture in each land-use tile[57]. An important feature of LM3 is that each land-use tile has its own soil water and plant root distribution: thus, evapotranspiration is not a function of the grid-cell average soil moisture. Observational analyses indicate that this distinction may be important. In particular, ref. [9] showed that the rate of consumption of root-zone water by plants drastically impacts heat wave severity, e.g., crops tend to transpire at peak capacity as long as water is available, while forests slow transpiration rates if temperatures get too high[9]. Models with one grid-cell average soil moisture reservoir will allow crops to continue transpiring by tapping into water that should be beyond the reach of crop roots, generating an unrealistically cooler and more humid response to crop cover.

Here, we compare the historical (i.e., 1861–2005) all-forcing simulation (AllHist) with simulations that do not include any anthropogenic interference through LULCC, referred to as the potential vegetation (PotVeg) experiment. All other radiative forcings in PotVeg are identical to those in the AllHist experiment. In both runs, the atmospheric $CO_2$ seen by the model's radiation code was restored to the observed historical trend on a 1-year time scale. Two ensemble members of each experiment were analyzed, with the last 25 years of each ensemble member (1981–2005) included in the analysis. Comparison of differences at the end of the two simulations captures the effects of the representation of the total anthropogenic signal of LULCC in the reconstruction[41]. Additional analyses of the first 25 years of the experiments (1861–1885) are presented in the Supplementary Information file to address differences that stem from LULCC that occurred prior to 1861. Differences between the two sets of figures will be discussed in the results section in order to identify the effects associated with LULCC over the more recent historical period.

**Analysis methods**. We use mixing diagrams following refs [58, 59] to highlight changes in the climatology and diurnal cycles induced by LULCC. The conserved variable mixing diagrams of Fig. 5 depict a measure of humidity ($\lambda q$) on the x-axis and a measure of temperature ($c_p T$) on the y-axis in common units of kJ kg$^{-1}$, where $\lambda$ is the latent heat of condensation, $c_p$ is the specific heat of dry air at constant pressure and $q$ and $T$ are 2-meter specific humidity and temperature. The joint temperature-humidity differences presented in Fig. 4 reflect the change in the mean conditions. To quantify the change in the occurrence of hot, dry extremes, we compute the amount of data from each experiment occupying the upper left portion of the mixing diagrams in Fig. 5. For convenience, we further define a set of orthogonal axes using the moist enthalpy, ME = $\lambda q + c_p T$, and a hot-dry measure, HD = $c_p T - \lambda q$. ME is a measure of the total internal energy of moist air[32], and is maximized in the upper right portion of the mixing diagrams. HD is maximized in the upper left corner of the mixing diagrams where conditions are both hot and dry. We use HD as a proxy for aridity; it also corresponds, qualitatively, to the behavior of saturation deficit, in the sense that high values of HD reflect larger saturation deficits. We calculate the 90th percentile threshold values for the monthly mean ME and HD. These threshold values are denoted in Fig. 5c with solid green (PotVeg) and black (AllHist) thin straight lines. Exceedances of these threshold values are calculated at each grid point, and the return periods associated with these exceedance rates are shown in Fig. 6a, c for individual months, with significance determined using a Wilcoxon rank sum test at the 5% significance level. Seasonal, latitudinal means of these return periods are shown in Fig. 6b, d. Our results do not change appreciably if we use wet-bulb temperature and RH instead of ME and HD (compare Fig. 6 with Supplementary Fig. 5), though the orthogonality and independence of the two measures is lost.

**Data availability**. Model source code is available at http://www.gfdl.noaa.gov/earth-system-model.

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

## Acknowledgements

We would like to thank Ramdane Alkama and Alessandro Cescatti for sharing observational data included in Fig. 2. We also thank Chris Milly and Tom Knutson for providing helpful comments on the manuscript.

## Author contributions

K.L.F. designed the study, analyzed the model output, and wrote the paper. J.P.K. performed the simulations. E.S. and S.M. provided important insight on aspects of the land and ESM models. J.A.S. provided important insight on the use of mixing diagrams. E.S., S.M., J.P.K., A.B., B.R.L. and P.G. contributed to the study design and analysis of results. All authors contributed to the writing and revising of the manuscript.

## Additional information

**Competing interests:** The authors declare no competing financial interests.

