## [Peer Review File · Nature Communications]

Reviewers' comments:

Reviewer #1 (Remarks to the Author):

Review of "The impact of historical land use/land cover change on regional climate extremes" by Findell et al.

The manuscript by Findell et al. addresses the important issue of the impacts of land cover changes on climate and climate extremes. This is a timely topic for which Nature Communications would potentially be a good outlet. The paper is generally well-written. In spite of the complexities involved in the modeling and the important advances made in including net and gross land cover changes in ESMs, the research itself is essentially a large sensitivity experiment in which two runs each consisting of 2 ensemble members are compared. Studies on the effect of land use changes on climate are fairly numerous, so the current manuscript should contain sufficient novel elements either in the experimental setup, analysis or insight, to merit publication. In my view, however, the current manuscript fails to convince that the results are sufficiently novel and in line with other (observational) evidence. As a result, I think major revisions are necessary before the manuscript can be considered for publication in Nature Communications. I will explain my reservations with the current version of the manuscript in more detail below.

In their introduction, the authors refer to the work by Pitman et al. (2012) to argue that individual models often produce opposing signs of LULCC. Given these results, it seems somewhat surprising that the authors have put little effort into showing the validity of their model in simulating continental-scale patterns of water and energy exchange, but in particular in simulating land-use specific behavior. Numerous datasets exist nowadays that can readily be used for validation, for instance FLUXNET or derived products such as the one by Martin Jung. Since the authors rely on a single model for their conclusions, detailed (land use specific) validation is essential and should be part of the manuscript (or SI).

A second concern deals with the mismatch between the experimental setup and the analysis that is performed on the model results. Simulations were performed for the period 1860-2005, yet only the period 1981-2005 seems to be analysed. This raises the question whether the experiments were actually designed for this study, and why no results are shown for earlier periods. Based on the land cover maps in Fig 1, it seems that most of the effect might already be present in the late 1800's (also due to potential inconsistencies between potential and 1860's vegetation, see next point), but this is not shown. Analysis over the whole simulation period is needed for a consistent story. Another aspect that is missing in the work is a clear hypothesis. Two simulations are performed that lead to different results, but what do we learn from this? How large are these differences when compared for instance to the effect of global warming? Instead of using two ensemble members, the authors could have chosen to perform two additional runs without CO2 increase, so that the effect of land use could have been separated from greenhouse effects. An important conclusion from the study is now that patterns of harvesting and regrowth are key to simulating changes, but the study has not been designed to isolate these effects since a comparison is made to potential vegetation and not the same land use but without gross changes. It is also not clear why the two ensemble members were used, since no statistics on robustness of the results are presented (an important weakness) that be affected by the number of simulation years. Overall, the study makes poor use of the available data, and main conclusions are drawn that are not in line with the experimental setup.

Third, the manuscript lacks a clear definition of "historical". By comparing potential vegetation (a model product) with observed land use changes, the authors implicitly consider changes over a much longer timeperiod than for which simulations have been performed. A relevant question is whether there has been any time during which vegetation has been "potential". To my best knowledge, land cover is also strongly controlled by factors such as (natural) grazing and fire. By not considering these factors, or by not attempting to validate the potential vegetation map, the

study is in effect an idealized comparison between forested and non-forested continents, and as a result the resulting signals will be overestimated. Since a significant amount of land use change has taken place in the period of simulation, the authors could have started the reference run with the (fixed) reconstructed 1860 land use rather than potential vegetation. This would have given a much more realistic (but likely smaller) signal. At least the consistency between the 1860 land use and potential vegetation should have been discussed.

Notwithstanding these comments, I believe a revised version could make a strong contribution to Nature Communications, and I hope the authors are able to address my concerns in a revision.

Reviewer #2 (Remarks to the Author):

This is a very important and clearly written paper. It will have a significant influence (or at least it should) on adequately assessing the role of humans on the climate system.

These findings from their study

Within GFDL EMS2G, the inclusion of anthropogenic LULCC produces a two-to-four-fold increase in the frequency of hot, dry summers in much of the mid-latitudes, as well as a two-to-four-fold increase of high humidity conditions throughout the tropics during all seasons, compared to what is seen with potential vegetation.

Shevliakova et al. (2009) and Houghton et al. (2012) showed that these practices impact carbon dynamics; here we demonstrate that they also impact the tropical hydrologic cycle.

These substantial and significant impacts call attention to the critical need to include land use-related processes in future projections of climate

reinforce the increasingly better recognized role of land use change and land management on the climate system.

The authors, however, leave out important previous papers on this subject. First, in their sentence

"... the novel consideration of the joint temperature–humidity response to historical LULCC"

using moist enthalpy, this valuable climate metric has already been presented for the same reasons as urged by the authors of this paper. This approach is discussed in these papers, for example,

Pielke Sr., R.A., C. Davey, and J. Morgan, 2004: Assessing "global warming" with surface heat content. *Eos*, 85, No. 21, 210-211

Davey, C.A., R.A. Pielke Sr., and K.P. Gallo, 2006: Differences between near-surface equivalent temperature and temperature trends for the eastern United States - Equivalent temperature as an alternative measure of heat content. *Global and Planetary Change*, 54, 19–32.

Fall, S., N. Diffenbaugh, D. Niyogi, R.A. Pielke Sr., and G. Rochon, 2010: Temperature and equivalent temperature over the United States (1979 – 2005). *Int. J. Climatol.*, DOI: 10.1002/joc.2094.

Peterson, T. C., K. M. Willett, and P. W. Thorne (2011), Observed changes in surface atmospheric energy over land, *Geophys. Res. Lett.*, 38, L16707, doi:10.1029/2011GL048442

Secondly, the urging of policymakers and the scientific community to include land-related processes beyond the carbon cycle, has been proposed in previous papers. These include

Marland, G., R.A. Pielke, Sr., M. Apps, R. Avissar, R.A. Betts, K.J. Davis, P.C. Frumhoff, S.T. Jackson, L. Joyce, P. Kauppi, J. Katzenberger, K.G. MacDicken, R. Neilson, J.O. Nilsson, D. Dutta S. Niyogi, R.J. Norby, N. Pena, N. Sampson, and Y. Xue, 2003: The climatic impacts of land surface change and carbon management, and the implications for climate-change mitigation policy. *Climate Policy*, 3, 149-157

Pielke Sr., R.A., A. Pitman, D. Niyogi, R. Mahmood, C. McAlpine, F. Hossain, K. Goldewijk, U. Nair, R. Betts, S. Fall, M. Reichstein, P. Kabat, and N. de Noblet-Ducoudré, 2011: Land use/land cover changes

and climate: Modeling analysis and observational evidence. *WIREs Clim Change* 2011, 2:828–850. doi: 10.1002/wcc.144

Mahmood, R., R.A. Pielke Sr., K. Hubbard, D. Niyogi, P. Dirmeyer, C. McAlpine, A. Carleton, R. Hale, S. Gameda, A. Beltrán-Przekurat, B. Baker, R. McNider, D. Legates, J. Shepherd, J. Du, P. Blanken, O. Frauenfeld, U. Nair, S. Fall, 2013: Land cover changes and their biogeophysical effects on climate. *Int. J. Climatol.*, DOI: 10.1002/joc.3736.

Mahmood, R., R.A. Pielke Sr., T.R. Loveland, and C.A. McAlpine, 2015: Climate relevant land use and land cover change policies. *Bull. Amer. Meteor. Soc.*, e-View doi: <http://dx.doi.org/10.1175/BAMS-D-14-00221.1>

Pielke Sr., R.A., R. Mahmood, and C. McAlpine, 2016: Land's complex role in climate change. *Physics Today*, 69(11), 40.

In terms of extreme extremes due to land use change and land management, the papers below provide examples of where this has been studied on the regional scale.

Hossain, F., I. Jeyachandran, and R.A. Pielke Sr., 2010: Dam safety effects due to human alteration of extreme precipitation. *Water Resources Research*, 46, W03301, doi:10.1029/2009WR007704.

Woldemichael, A., F. Hossain, R.A. Pielke Sr., and A. Beltrán-Przekurat, 2012: Understanding the impact of dam-triggered land use/land cover change on the modification of extreme precipitation, *Water Resour. Res.*, doi:10.1029/2011WR011684.

Hossain, F., J. Arnold, E. Beighley, C. Brown, S. Burian, J. Chen, S. Madadgar, A. Mitra, D. Niyogi, R.A. Pielke Sr., V. Tidwell, and D. Wegner, 2015: Local-to-regional landscape drivers of extreme weather and climate: Implications for water infrastructure resilience. *J. Hydrol. Eng.*, 10.1061/(ASCE)HE.1943-5584.0001210 , 02515002.

The authors build further on such past work.

The authors, also, need to modify the following sentence

The atmosphere and land components' resolution is 2° latitude by 2.5° longitude, with 24 vertical levels in the atmosphere and 20 layers in the soil.

It should read

The atmosphere and land components' HORIZONTAL GRID INCREMENT ARE 2° latitude by 2.5° longitude, with 24 vertical levels in the atmosphere and 20 layers in the soil.

At least 4 grid increments are needed to resolve a feature in a numerical model. This mean the actual spatial resolution is much coarser. The authors should urge for higher spatial resolution runs as the spatial scale of land use is on smaller scales. Indeed, this is a critical requirement for follow on work to assess if their findings are robust as finer resolution land structure is assessed. This could be done with regional models as exemplified in several of the papers listed above.

With the authors considering this comments in this review, I recommend the paper be accepted. It will be a noteworthy and important contribution to climate science and will be of wide interest.

Reviewer #3 (Remarks to Author)

Review of the paper entitled

The impact of historical land use/land cover change on regional climate extremes’ from Findell et al.

The paper proposed by Findell and coF authors addresses a very important issue, insufficiently covered by our scientific literature: the impact of changes in land cover (LCC) on extreme weather variables. There is today much information regarding the impact of LCC on mean climatology, at both the regional and global scales, but there is less analysis devoted to changes in variability and extremes.

In that sense **this paper brings some novelty in this field of science.**

It shows that humanF induced land cover changes, since humans have started to modify the landscape, has essentially increased warm and arid events in temperate regions, humid ones in the tropics. They have calculated the number of years, out of 10, that now experience some conditions that were rarely reached when vegetation was pristine (events that would occur less than 1 out of 10 years).

Simulations and results are clear, well presented, although I have some comments on those that I will describe further. The methods used for analysis would deserve some clarifications. Figures are selfF sufficient and consistently support the discussion in the text. I have however some questions, doubts about the results obtained for the tropics that I will get into later on.

Having said that I think **this paper deserves publication albeit after some corrections** that I call major to make sure they will be done but that should not take too much time to the authors. There is some clarification to be made in the text.

Specific comments:

• **Interpretation of the title**

- The title refers to ‘historical’ changes in land cover. This is in general interpreted as ‘since pre-industrial times’ in the climate community, while you refer to more drastic changes: from pristine to today’s vegetation distribution. You may want to clarify that in the title as well as in the abstract.

• **The simulation set-up**

- I agree that your land-surface model is probably at the ‘forefront of process-inclusion ... regarding land use ..’. However I doubt you had time to evaluate it against Alkama & Cescatti (2016) datasets. Or if you have and if such paper is not published yet it would be nice to show in the supplementary material section proofs of the ‘good evaluation’ you claim.
- I have trouble understanding how you can have only 5 vegetation types within which you distinguish sub-layers such as crops, pastures, ... I admit I did not read the paper describing your model and probably others will not SO it would be nice to clarify that point.
- It is not clear to me what you do with SSTs for both 25 years ensemble members. I imagine you prescribe them from the AllHist simulations so that all members and ensembles share the exact same SSTs but ... I cannot find the info in the paper. Would you please clarify this?
- Could you comment on the number of years available for your analysis? Are 2 members per ensemble sufficient?

• **The analysis methods**

- I feel you are going quite fast in the description of your analysis method ... what λ_q and $c_p T$ mean in physical terms for a reader that would be non specialist may be hard to understand so I would recommend to take some time to improve the explanations. This could potentially be done in the supplementary section.

- I'm not sure I understand why orthogonal to constant ME lines provide infos on aridity ... here again this may deserve some clarification.
 - The way the return periods are being calculated may deserve clarification as well.
- **Results for northern mid latitudes**
 - I think there is a mistake in the left panel of Figure 3 as X & Y axis should show $\Delta\lambda_q$ and $\Delta c_p T$ (i.e. anomalies) rather than λ_q and $c_p T$ (values for a specific simulation).
 - You do not discuss in the text the fact that those anomalies (Δ) are very small with respect to absolute changes: $\Delta c_p T \sim 0,5\%$ and $\Delta\lambda_q \leq 5\%$.. the choice of color you made is very convincing but values are in reality very small I think you should not discard/bury that, as recognizing that changes in mean values may seem negligible gives even more weight to the fact that such diagnostics are not the right ones!
 - Your interpretation in the text in terms of N years out of 10 that go beyond some thresholds is difficult to follow for me as I'm not 100% sure how you calculated those numbers. From the text it seems that you derive this from looking at Figure 4 but then you show numbers on a map in Figure 5 ... so once again I think this would deserve clarification on the way you make those calculations.
 - Moreover in the text it seems that those return values are calculated from averaged daily cycles while they should be calculated from all available outputs. Could you please clarify that?
 - **Results for the tropics**
 - Regarding the tropics it is not at all clear to me why GPP increases everywhere even in large areas where LAI decreases.
 - It is not clear either why for example LAI increases in northern south America (central part) while there is no apparent vegetation change in Figure 1.
 - As most interpretations of changes in moisture derive from the change in GPP I'm not confident about the results you show and your discussion for this part of the world. I would recommend you start explaining the changes you simulate ... they may just be noise don't you think?

I hope those comments will help the authors to improve their manuscript.

I'm signing my review

- Nathalie de Noblet-Ducoudré

Responses to Reviews of “The Impact of Anthropogenic Land Use and Land Cover Change on Regional Climate Extremes” by Findell et al.

We are grateful to the three reviewers for their helpful suggestions and for their positive responses to our paper. We have responded to each specific suggestion, and we feel our manuscript is substantially stronger as a result of these improvements. We have made a number of broad changes to the manuscript to address the largest concerns:

1. We added a new figure (now Figure 2) which compares model output with data from Alkama and Cescatti’s 2016 paper. We believe this demonstrates that the model’s mean temperature response to deforestation is consistent with observations in boreal, temperate, and tropical ecosystems. We feel that this additional figure substantially increases the potential impact of the paper.
2. We have included supplementary figures analyzing the differences between the AllHist and PotVeg runs at the beginning of the simulations (1861-1885).
3. We have more carefully and clearly used the terms ‘historical’ and ‘anthropogenic’ throughout the paper.
4. We have expanded our discussion of the statistical techniques we employ and clarified their usage when the results are presented. All the figures which relied on t-tests now use the modified version, accounting for autocorrelation within the time series.
5. We have changed the discussion of the tropical results to reflect the fact that these signals are not consistent with the tropical response in Malyshev et al (2015), and in fact point to the likelihood that these simulations undersample the long time-scale variability of the tropical ocean.
6. We have added many important additional references.

Our responses to the reviewers’ individual comments are listed below in blue italicized font.

Reviewer #1’s comments

Review of “The impact of historical land use/land cover change on regional climate extremes” by Findell et al.

The manuscript by Findell et al. addresses the important issue of the impacts of land cover changes on climate and climate extremes. This is a timely topic for which Nature Communications would potentially be a good outlet. The paper is generally well-written. In spite of the complexities involved in the modeling and the important advances made in including net and gross land cover changes in ESMs, the research itself is essentially a large sensitivity experiment in which two runs each consisting of 2 ensemble members are compared. Studies on the effect of land use changes on climate are fairly numerous, so the current manuscript should contain sufficient novel elements either in the experimental setup, analysis or insight, to merit publication. In my view, however, the current manuscript fails to convince that the results are sufficiently novel and in line with other (observational) evidence. As a result, I think major

revisions are necessary before the manuscript can be considered for publication in Nature Communications. I will explain my reservations with the current version of the manuscript in more detail below.

In their introduction, the authors refer to the work by Pitman et al. (2012) to argue that individual models often produce opposing signs of LULCC. Given these results, it seems somewhat surprising that the authors have put little effort into showing the validity of their model in simulating continental-scale patterns of water and energy exchange, but in particular in simulating land-use specific behavior. Numerous datasets exist nowadays that can readily be used for validation, for instance FLUXNET or derived products such as the one by Martin Jung. Since the authors rely on a single model for their conclusions, detailed (land use specific) validation is essential and should be part of the manuscript (or SI).

Response: Results presented in the new Figure 2, alongside observations from Alkama and Cescatti (2016), confirm that the behavior of the GFDL land model in response to deforestation is in-line with observations. For summer in the midlatitudes, this is in contrast with the majority of land surface models included in studies such as Pitman et al. (2009). We are pleased that the reviewers emphasized the importance of this comparison—the paper is much stronger with this observational validation.

A second concern deals with the mismatch between the experimental setup and the analysis that is performed on the model results. Simulations were performed for the period 1860-2005, yet only the period 1981-2005 seems to be analysed. This raises the question whether the experiments were actually designed for this study, and why no results are shown for earlier periods. Based on the land cover maps in Fig 1, it seems that most of the effect might already be present in the late 1800's (also due to potential inconsistencies between potential and 1860's vegetation, see next point), but this is not shown. Analysis over the whole simulation period is needed for a consistent story.

Response: Indeed, the experiments used in this analysis were designed for other studies, and yet they presented us with a wonderful opportunity to learn something about changes in extremes in response to LULCC. We have now included Supplementary Figures comparing behavior of regional extremes in the PotVeg and AllHist experiments at the beginning of the simulations (1861-1885). These new supplemental figures are discussed primarily at the end of Section 3.1.

Another aspect that is missing in the work is a clear hypothesis. Two simulations are performed that lead to different results, but what do we learn from this? How large are these differences when compared for instance to the effect of global warming? Instead of using two ensemble members, the authors could have chosen to perform two additional runs without CO₂ increase, so that the effect of land use could have been separated from greenhouse effects. An important conclusion from the study is now that patterns of harvesting and regrowth are key to simulating changes, but the study has not been designed to isolate these effects since a comparison is made to potential vegetation and not the same land use but without gross changes.

***Response:** In a 2009 paper with an earlier generation GFDL model (Findell et al., 2009) we compared the magnitude of regional signals of LULCC to the magnitude of greenhouse warming signal. That paper concluded that in altered regions, LULCC impacts were as important as impacts from greenhouse warming. The magnitude of the LULCC signal in midlatitudes is similar in this generation model. We now specifically mention this result to highlight the comparable importance of these impacts.*

It is also not clear why the two ensemble members were used, since no statistics on robustness of the results are presented (an important weakness) that be affected by the number of simulation years. Overall, the study makes poor use of the available data, and main conclusions are drawn that are not in line with the experimental setup.

***Response:** Our statistical techniques are detailed in the Methods section, but the earlier draft failed to bring attention to this when the figures and results were presented. We now emphasize that the results presented are indeed subjected to statistical analysis when we discuss each figure. We also updated many figures so that all the figures which rely on t-tests are computed with the modified t-test, accounting for autocorrelation within the time series. We also discovered that the modified t-test had a bug which was erroneously masking out negative differences in addition to non-significant differences. This bug masked the signal in the North Atlantic in the previous version of Figure 4. We now include a discussion of this region in Section 3.1.*

Third, the manuscript lacks a clear definition of “historical”. By comparing potential vegetation (a model product) with observed land use changes, the authors implicitly consider changes over a much longer time period than for which simulations have been performed. A relevant question is whether there has been any time during which vegetation has been “potential”. To my best knowledge, land cover is also strongly controlled by factors such as (natural) grazing and fire. By not considering these factors, or by not attempting to validate the potential vegetation map, the study is in effect an idealized comparison between forested and non-forested continents, and as a result the resulting signals will be overestimated. Since a significant amount of land use change has taken place in the period of simulation, the authors could have started the reference run with the (fixed) reconstructed 1860 land use rather than potential vegetation. This would have given a much more realistic (but likely smaller) signal. At least the consistency between the 1860 land use and potential vegetation should have been discussed.

***Response:** We changed the title to use the word ‘anthropogenic’ rather than ‘historical.’ In the earlier manuscript we used these words interchangeably: here we are more precise. Additionally, through the added presentation of supplemental figures analyzing the differences between the AllHist and PotVeg runs at the beginning of the simulations (1861-1885), we can now make a distinction between the changes that happened before and after the late 1800s. As we discuss in Section 3.1, minor differences were present in the late 1800s, but the magnitude, significance and spatial extent of the differences in the means is far larger in the more recent period, and the changes in the extremes are largely absent in the figures for the late 1800s, indicating that most of the changes to the tails of the distribution has*

occurred since 1885.

Notwithstanding these comments, I believe a revised version could make a strong contribution to Nature Communications, and I hope the authors are able to address my concerns in a revision.

Response: *We too hope that we have fully addressed your concerns!*

Reviewer #2's comments

This is a very important and clearly written paper. It will have a significant influence (or at least it should) on adequately assessing the role of humans on the climate system.

These findings from their study

Within GFDL EMS2G, the inclusion of anthropogenic LULCC produces a two-to-four-fold increase in the frequency of hot, dry summers in much of the mid-latitudes, as well as a two-to-four-fold increase of high humidity conditions throughout the tropics during all seasons, compared to what is seen with potential vegetation.

Shevliakova et al. (2009) and Houghton et al. (2012) showed that these practices impact carbon dynamics; here we demonstrate that they also impact the tropical hydrologic cycle.

These substantial and significant impacts call attention to the critical need to include land use-related processes in future projections of climate

reinforce the increasingly better recognized of the role of land use change and land management on the climate system.

The authors, however, leave out important previous papers on this subject. First, in their sentence

"... the novel consideration of the joint temperature–humidity response to historical LULCC"

using moist enthalpy, this valuable climate metric has already been presented for the same reasons as urged by the authors of this paper. This approach is discussed in these papers, for example,

Pielke Sr., R.A., C. Davey, and J. Morgan, 2004: Assessing "global warming" with surface heat content. *Eos*, 85, No. 21, 210-211

Davey, C.A., R.A. Pielke Sr., and K.P. Gallo, 2006: Differences between near-surface equivalent temperature and temperature trends for the eastern United States - Equivalent temperature as an alternative measure of heat content. *Global and Planetary Change*, 54, 19–32.

Fall, S., N. Diffenbaugh, D. Niyogi, R.A. Pielke Sr., and G. Rochon, 2010: Temperature and equivalent temperature over the United States (1979 – 2005). *Int. J. Climatol.*, DOI: 10.1002/joc.2094.

Peterson, T. C., K. M. Willett, and P. W. Thorne (2011), Observed changes in surface atmospheric energy over land, *Geophys. Res. Lett.*, 38, L16707, doi:10.1029/2011GL048442

Response: We appreciate the reviewer's comments and are grateful for the reminder of these important references. We have added mention of Fall et al. (2010), Pielke et al. (2004), and Davey et al. (2006) in multiple locations in the manuscript. They are highly relevant to this study.

Secondly, the urging of policymakers and the scientific community to include land-related processes beyond the carbon cycle, has been proposed in previous papers. These include

Marland, G., R.A. Pielke, Sr., M. Apps, R. Avissar, R.A. Betts, K.J. Davis, P.C. Frumhoff, S.T. Jackson, L. Joyce, P. Kauppi, J. Katzenberger, K.G. MacDicken, R. Neilson, J.O. Niles, D. Dutta S. Niyogi, R.J. Norby, N.

Pena, N. Sampson, and Y. Xue, 2003: The climatic impacts of land surface change and carbon management, and the implications for climate-change mitigation policy. *Climate Policy*, 3, 149-157

Pielke Sr., R.A., A. Pitman, D. Niyogi, R. Mahmood, C. McAlpine, F. Hossain, K. Goldewijk, U. Nair, R. Betts, S. Fall, M. Reichstein, P. Kabat, and N. de Noblet-Ducoudré, 2011: Land use/land cover changes and climate: Modeling analysis and observational evidence. *WIREs Clim Change* 2011, 2:828–850. doi: 10.1002/wcc.144

Mahmood, R., R.A. Pielke Sr., K. Hubbard, D. Niyogi, P. Dirmeyer, C. McAlpine, A. Carleton, R. Hale, S. Gameda, A. Beltrán-Przekurat, B. Baker, R. McNider, D. Legates, J. Shepherd, J. Du, P. Blanken, O. Frauenfeld, U. Nair, S. Fall, 2013: Land cover changes and their biogeophysical effects on climate. *Int. J. Climatol.*, DOI: 10.1002/joc.3736.

Mahmood, R., R.A. Pielke Sr., T.R. Loveland, and C.A. McAlpine, 2015: Climate relevant land use and land cover change policies. *Bull. Amer. Meteor. Soc.*, e-View doi: <http://dx.doi.org/10.1175/BAMS-D-14-00221.1>

Pielke Sr., R.A., R. Mahmood, and C. McAlpine, 2016: Land's complex role in climate change. *Physics Today*, 69(11), 40.

Response: We have added mention of the highly-relevant Pielke et al. (2011) and Mahmood et al. (2013) papers. We did not feel the policy-relevant papers were as closely related to the issues we raise, though they are important additions to the literature.

In terms of extreme extremes due to land use change and land management, the papers below provide examples of where this has been studied on the regional scale.

Hossain, F., I. Jeyachandran, and R.A. Pielke Sr., 2010: Dam safety effects due to human alteration of extreme precipitation. *Water Resources Research*, 46, W03301, doi:10.1029/2009WR007704.

Woldemichael, A., F. Hossain, R.A. Pielke Sr., and A. Beltrán-Przekurat, 2012: Understanding the impact of dam-triggered land use/land cover change on the modification of extreme precipitation, *Water Resour. Res.*, doi:10. 1029/ 2011 WR011684.

Hossain, F., J. Arnold, E. Beighley, C. Brown, S. Burian, J. Chen, S. Madadgar, A. Mitra, D. Niyogi, R.A. Pielke Sr., V. Tidwell, and D. Wegner, 2015: Local-to-regional landscape drivers of extreme weather and climate: Implications for water infrastructure resilience. *J. Hydrol. Eng.*, 10.1061/(ASCE)HE.1943-5584.0001210 , 02515002.

The authors build further on such past work.

Response: These are interesting papers that we were happy to read. However, given that they focus on infrastructure resilience to precipitation extremes, we felt they were not as relevant to our manuscript as many of the other references the reviewer suggested.

The authors, also, need to modify the following sentence

The atmosphere and land components' resolution is 2° latitude by 2.5° longitude, with 24 vertical levels in the atmosphere and 20 layers in the soil.

It should read

The atmosphere and land components' HORIZONTAL GRID INCREMENT ARE 2° latitude by 2.5° longitude, with 24 vertical levels in the atmosphere and 20 layers in the soil.

Response: The sentence has been modified. Thank you.

At least 4 grid increments are needed to resolve a feature in a numerical model. This mean the actual spatial resolution is much coarser. The authors should urge for higher spatial resolution runs as the spatial scale of land use is on smaller scales. Indeed, this is a critical requirement for follow on work to assess if their findings are robust as finer resolution land structure is assessed. This could be done with regional models as exemplified in several of the papers listed above.

Response: We have changed the final sentence of the paper to stress the need for high resolution representations of LULCC.

With the authors considering this comments in this review, I recommend the paper be accepted. It will be a noteworthy and important contribution to climate science and will be of wide interest.

Reviewer #3's comments

Review of the paper entitled The impact of historical land use/land cover change on regional climate extremes

from Findell et al.

The paper proposed by Findell and co-authors addresses a very important issue, insufficiently covered by our scientific literature: the impact of changes in land cover (LCC) on extreme weather variables. There is today much information regarding the impact of LCC on mean climatology, at both the regional and global scales, but there is less analysis devoted to changes in variability and extremes.

In that sense **this paper brings some novelty in this field of science.**

It shows that human-induced land cover changes, since humans have started to modify the landscape, has essentially increased warm and arid events in temperate regions, humid ones in the tropics. They have calculated the number of years, out of 10, that now experience some conditions that were rarely reached when vegetation was pristine (events that would occur less than 1 out of 10 years).

Simulations and results are clear, well presented, although I have some comments on those that I will describe further. The methods used for analysis would deserve some clarifications. Figures are self-sufficient and consistently support the discussion in the text. I have however some questions, doubts about the results obtained for the tropics that I will get into later on.

Having said that I think **this paper deserves publication albeit after some corrections** that I call major to make sure they will be done but that should not take too much time to the authors. There is some clarification to be made in the text.

Specific comments:

- **Interpretation of the title**

- The title refers to 'historical' changes in land cover. This is in general interpreted as 'since pre-industrial times' in the climate community, while you refer to more drastic changes: from pristine to today's vegetation distribution. You may want to clarify that in the title as well as in the abstract.

Response: We have changed the title (and the text) to use the word anthropogenic rather than historical. We now only use historical when discussing the differences between the figures in the main body of the manuscript and those in the supplement which focus on AllHist-PotVeg from the end of the 1800s.

- **The simulation set-up**

- I agree that your land-surface model is probably at the 'forefront of process-

inclusion ... regarding land use ..'. However I doubt you had time to evaluate it against Alkama & Cescatti (2016) datasets. Or if you have and if such paper is not published yet it would be nice to show in the supplementary material section proofs of the 'good evaluation' you claim.

Response: As mentioned in our list of broad changes and in response to Reviewer 1, we have a new figure directly comparing the model results to the Alkama and Cescatti (2016) dataset. We believe this new figure makes for a much stronger paper.

- I have trouble understanding how you can have only 5 vegetation types within which you distinguish sub-layers such as crops, pastures, ... I admit I did not read the paper describing your model and probably others will not SO it would be nice to clarify that point.

Response: We have included the following sentence to clarify this aspect of the model: "The biogeography parameterization uses the total biomass in a tile in combination with prevailing climatic conditions to determine the vegetation type."

- It is not clear to me what you do with SSTs for both 25 years ensemble members. I imagine you prescribe them from the AllHist simulations so that all members and ensembles share the exact same SSTs but ... I cannot find the info in the paper. Would you please clarify this?

Response: The model has a fully interactive ocean. This is now mentioned explicitly in the model description section (Section 2.1).

- Could you comment on the number of years available for your analysis? Are 2 members per ensemble sufficient?

Response: Our statistics are applied to 50 years of data. For the mid-latitude land results, we believe this is enough to be representative of conditions at the end of the 20th Century. Our statistical techniques take into account the number of years, and results are only shown where they pass the statistical techniques described. Our initial analyses were on the last 20 years of each simulation (40 years in total), but we expanded the time period before finalizing the manuscript. The conclusions did not change as a result of the addition of these 10 years.

However, this comment and subsequent comments from the reviewer did prompt us to look in more detail at the variability of the ocean in these simulations and in the simulations Sergey Malyshev ran for his 2015 paper using the model ESM2Mb. This model differs from ESM2G only in the ocean component, and yet the tropical response in that model is indeed different. We have modified our discussion of and conclusions about the tropics to reflect the need for larger ensembles and/or longer simulations to adequately sample the tropical variability imposed by the long time scales of the ocean.

• **The analysis methods**

- I feel you are going quite fast in the description of your analysis method ... what λ_q and c_pT mean in physical terms for a reader that would be non-specialist may be hard to understand so I would recommend to take some time to improve the explanations. This could potentially be done in the supplementary section.
- I'm not sure I understand why orthogonal to constant ME lines provide infos on

aridity ... here again this may deserve some clarification.

- The way the return periods are being calculated may deserve clarification as well.

Response: These points are well-taken. We have expanded the explanations of the methods quite considerably in the hopes that non-specialists can follow our approach. In particular, descriptions of mixing diagrams and the return period methodology has been expanded, and the lines orthogonal to constant ME have been named (HD, see Section 2.3) so that they are easier to discuss and describe, and the connection to aridity can be made clearer.

- **Results for northern mid latitudes**

- I think there is a mistake in the left panel of Figure 3 as X & Y axis should show $\Delta\lambda_q$ and $\Delta c_p T$ (i.e. anomalies) rather than λ_q and $c_p T$ (values for a specific simulation).

Response: Thanks for catching this. The axis labels have been corrected.

- You do not discuss in the text the fact that those anomalies (Δ) are very small with respect to absolute changes: $\Delta c_p T \sim 0.5\%$ and $\Delta\lambda_q \leq 5\%$.. the choice of color you made is very convincing but values are in reality very small I think you should not discard/bury that, as recognizing that changes in mean values may seem negligible gives even more weight to the fact that such diagnostics are not the right ones!

Response: The text has been changed to highlight these points, particularly in the middle of Section 3.1.

- Your interpretation in the text in terms of N years out of 10 that go beyond some thresholds is difficult to follow for me as I'm not 100% sure how you calculated those numbers. From the text it seems that you derive this from looking at Figure 4 but then you show numbers on a map in Figure 5 ... so once again I think this would deserve clarification on the way you make those calculations.

Response: This methodology is described in much more detail now.

- Moreover in the text it seems that those return values are calculated from averaged daily cycles while they should be calculated from all available outputs. Could you please clarify that?

Response: The data available for analysis are monthly mean diurnal cycles (24 hourly data points). Our discussion near the end of Section 3.1 was intended to point out that if hourly data were saved every day, there might be larger differences between the analysis of the daily means versus daily Tmax or 3-6pm averages. We have described these distinctions more carefully now.

- **Results for the tropics**

Response: As mentioned in our list of broad changes, the reviewer's inquiries about the tropics prompted us to look in detail at the tropical results shown in similar runs with the sister model ESM2Mb used in Malyshev et al. (2015). This led to a number of changes in the manuscript.

- Regarding the tropics it is not at all clear to me why GPP increases everywhere even in large areas where LAI decreases.

Response: We have expanded our discussion of the LAI changes in the tropics, and make a clearer distinction between areas with and without crops. The GPP increases are discussed in concert with increases in temperature and humidity (and ME) over the tropical oceans and, indeed, extending across most of the tropics. As is now mentioned, these and other factors preclude determination of how much of these tropical differences stem from the model's representation of land use practices and how much is forced by internal variability within the tropical oceans.

- It is not clear either why for example LAI increases in northern South America (central part) while there is no apparent vegetation change in Figure 1.

Response: This figure is now plotted with a modified t-test, and only isolated pixels with LAI increases are significant under this more rigorous test. However, as we discuss in Section 3.2, these LAI changes are not repeated in the runs of Malyshev et al. (2015). This and further analysis of the runs used in that 2015 paper indicate that these LAI changes are not robust.

- As most interpretations of changes in moisture derive from the change in GPP I'm not confident about the results you show and your discussion for this part of the world. I would recommend you start explaining the changes you simulate ... they may just be noise don't you think?

Response: As mentioned above, this comment prompted us to look in detail at the 5 ensemble members Sergey Malyshev ran with ESM2Mb to look at anthropogenic LULCC. Indeed, that model shows different results in the tropics than we show here. The text for this section is heavily altered, reflecting the likelihood that our two 25-year-long ensemble members are not sufficient to sample the full range of internal variability in the tropical ocean system. Thank you for prompting us to look at this issue more closely.

I hope those comments will help the authors to improve their manuscript. I'm signing my review - Nathalie de Noblet-Ducoudré

REVIEWERS' COMMENTS:

Reviewer #1 (Remarks to the Author):

The authors have adequately addressed my comments on the initial submission. I am pleased to see a direct evaluation of the land use effect in the model against observations in the new version. This makes the results much more convincing. I recommend to accept this version for publication.

Reviewer #2 (Remarks to the Author):

The authors satisfactorily addressed the review comments. I recommend acceptance.